# Low-shot Learning via Covariance-Preserving Adversarial Augmentation Networks

**Hang Gao**[1]**, Zheng Shou**[1]**, Alireza Zareian**[1]**, Hanwang Zhang**[2]**, Shih-Fu Chang**[1]
[1]Columbia University, [2]Nanyang Technological University
{hg2469, zs2262, az2407, sc250}@columbia.edu
hanwangzhang@ntu.edu.sg

## Abstract

Deep neural networks suffer from over-fitting and catastrophic forgetting when trained with small data. One natural remedy for this problem is data augmentation, which has been recently shown to be effective. However, previous works either assume that intra-class variances can always be generalized to new classes, or employ naive generation methods to hallucinate finite examples without modeling their latent distributions. In this work, we propose *Covariance-Preserving Adversarial Augmentation Networks* to overcome existing limits of low-shot learning. Specifically, a novel Generative Adversarial Network is designed to model the latent distribution of each novel class given its related base counterparts. Since direct estimation of novel classes can be inductively biased, we explicitly preserve covariance information as the "variability" of base examples during the generation process. Empirical results show that our model can generate realistic yet diverse examples, leading to substantial improvements on the ImageNet benchmark over the state of the art.

## 1 Introduction

The hallmark of learning new concepts from very few examples characterizes human intelligence. Though constantly pushing limits forward in various visual tasks, current deep learning approaches struggle in cases when abundant training data is impractical to gather. A straightforward idea to learn new concepts is to fine-tune a model pre-trained on *base* categories, using limited data from another set of *novel* categories. However, this usually leads to catastrophic forgetting [1], i.e., fine-tuning makes the model over-fitting on novel classes, and agnostic to the majority of base classes [2, 3], deteriorating overall performance.

One way to address this problem is to augment data for novel classes. Since generating images could be both unnecessary [4] and impractical [5] on large datasets, feature augmentation [6, 7] is more preferable in this scenario. Building upon learned representations [8, 9, 10], recently two variants of generative models show the promising capability of learning variation modes from base classes to imagine the missing pattern of novel classes. Hariharan *et al.* proposed *Feature Hallucination (FH)* [11], which can learn a finite set of transformation mappings between examples in each base category and directly apply them to seed novel points for extra data. However, since mappings are *enumerable* (even in large amount), this model suffers from poor generalization. To address this issue, Wang *et al.* [12] proposed *Feature Imagination (FI)*, a meta-learning based generation framework that can train an agent to synthesize extra data given a specific task. They circumvented the demand for latent distribution of novel classes by end-to-end optimization. But the generation results usually collapse into certain modes. Finally, it should be noted that both works erroneously assume that intra-class variances of base classes are shareable with any novel classes. For example, the visual variability of the concept *lemon* cannot be generalized to other irrelevant categories such as *raccoon*.

In this work, we propose a new approach to addressing the problem of low-shot learning by enabling better feature augmentation beyond current limits. Our approaches are novel in two aspects: modeling and training strategy. We propose *Covariance-Preserving Adversarial Augmentation Networks (CP-AAN)*, a new class of Generative Adversarial Networks (GAN) [14, 15] for feature augmentation. We take inspiration from unpaired image-to-image translation [16, 17] and formulate our feature augmentation problem as an imbalanced set-to-set translation problem where the conditional distribution of examples of each novel class can be conceptually expressed as a mixture of related base classes. We first extract all related base-novel class pairs by an intuitive yet effective approach called *Neighborhood Batch Sam-*

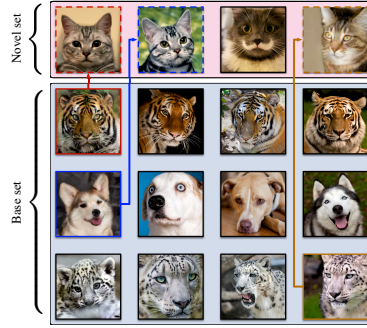

Figure 1: **Conceptual illustration of our method.** Given an example from a novel class, we translate examples from related base classes into the target class for augmentation. Image by [13].

*pling*. Then, our model aims to learn the latent distribution of each novel class given its base counterparts. Since the direct estimation of novel classes can be inductively biased during this process, we explicitly preserve the covariance base examples during the generation process.

We systematically evaluate our approach by considering a series of objective functions. Our model achieves the state-of-the-art performance over the challenging ImageNet benchmark [18]. With ablation studies, we also demonstrate the effectiveness of each component in our method.

## 2 Related Works

**Low-shot Learning**     For quick adaptation when very few novel examples are available, the community has often used a meta-agent [19] to further tune base classifiers [8, 9, 10]. Intuitive yet often ignored, feature augmentation was recently brought into the field by Hariharan *et al.* [11] to ease the data scarce scenario. Compared to traditional meta-learning based approaches, they have reported noticeable improvement on not only the *conventional* setting (i.e., to test on novel examples only), but also the more challenging *generalized* setting (i.e., to test on all classes). Yet the drawback is that both the original work and its variants [12] fail to synthesize diverse examples because of ill-constrained generation processes. Our approach falls in this line of research while seeking more principal guidance from base examples in a selective, class-specific manner.

**Generative Adversarial Network for Set-to-set Translation**     GANs [14] map each latent code from an easily sampled prior to a realistic sample of a complex target distribution. Zhu *et al.* [16] have achieved astounding results on image-to-image translation without any paired training samples. In our case, diverse feature augmentation is feasible through conditional translation given a pair of related novel and base classes. Yet two main challenges remain: practically, not all examples are semantically translatable. Second, given extremely scarce data for novel classes, we are unable to estimate their latent distributions (see Figure 4). In this work, we thoroughly investigate conditional GAN variants inspired by previous works [5, 15, 17, 20] to enable low-shot generation. Furthermore, we introduce a novel batch sampling technique for learning salient set-to-set mappings using unpaired data with categorical conditions.

**Generation from Limited Observations**     Estimation of latent distribution from a handful of observations is biased and inaccurate [21, 22]. The Bayesian approaches aim to model latent distributions of a variety of classes as hierarchical Gaussian mixture [23], or alternatively model generation as a sequential decision making process [24]. For GANs, Gaussian mixture noise has also been incorporated for latent code sampling [25]. Recent works [26, 27] on integral probability metrics provide theoretical guidance towards the high order feature matching. In this paper, building upon the assumption that related classes should have similar intra-class variance, we introduce a new loss term for preserving covariance during the translation process.

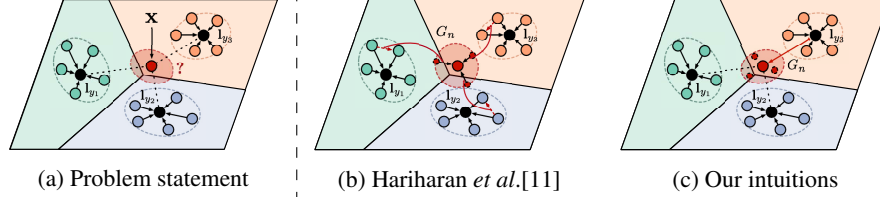

| (a) Problem statement | (b) Hariharan *et al.*[11] | (c) Our intuitions |

Figure 2: **Imbalanced set-to-set translation and our motivations.** Examples of three base classes are visualized in a semantic space learned by Prototypical Networks [9], along with their centroids as class prototypes. **(a):** Given a novel example $\mathbf{x}$, our goal is to translate base examples into the novel class, to reconstruct an estimation of the novel class distribution. **(b):** Feature Hallucination [11] randomly applies transformation mappings, between sampled base pairs in the same class, to the seed novel example for extra data; **(c):** instead, we only refer to semantically similar base-novel class pairs and model the distribution of data for novel classes by preserving base intra-class variances.

# 3 Imbalanced Set-to-set Translation

In this section, we formulate our low-shot feature augmentation problem under an imbalanced set-to-set translation framework. Concretely, we are given two labeled datasets represented in the same $D$-dimensional semantic space: (1) a base set $\mathcal{B} = \{(\mathbf{x}_b, y_b) \,|\, \mathbf{x}_b \in \mathbb{R}^D, y_b \in \mathcal{Y}_b\}$ consisting of abundant samples and (2) a novel set $\mathcal{N} = \{(\mathbf{x}_n, y_n) \,|\, \mathbf{x}_n \in \mathbb{R}^D, y_n \in \mathcal{Y}_n\}$ with only a handful of observations. Their discrete label spaces are assumed to be non-overlapping, i.e., $\mathcal{Y}_b \cap \mathcal{Y}_n = \varnothing$. Our goal is to learn a mapping function $G_n : \mathcal{B} \mapsto \mathcal{N}$ in order to translate examples of the base classes into novel categories. After the generation process, a final classifier is trained using both original examples of the base classes and all (mostly synthesized) examples of the novel classes.

Existing works [11, 12] suffer from the use of arbitrary, and thus possibly unrelated, base classes for feature augmentation. Moreover, their performances are degraded by naive generation methods without modeling the latent distribution of each novel class. Our insight, conversely, is to sample extra features from continuous latent *distributions* rather than certain modes from *enumerations*, by learning a GAN model (see Figure 2).

Specifically, we address two challenges that impede good translation under imbalanced scenarios: (1) through which base-novel class pairs we can translate; and more fundamentally, (2) through what objectives for GAN training we can estimate the latent distribution of novel classes with limited observations. We here start by proposing a straightforward batch sampling technique to address the first problem. Then we suggest a simple extension of existing methods and study its weakness, which motivates the development of our final approach. For clarity, we introduce a toy dataset for imbalanced set-to-set translation in Figure 3 as a conceptual demonstration of the proposed method compared to baselines.

## 3.1 Neighborhood Batch Sampling

It is widely acknowledged [28, 8, 9] that a metric-learned high dimensional space encodes relational semantics between examples. Therefore, to define which base classes are translatable to a novel class, we can rank them by their distance in a semantic space. For simplicity, we formulate our approach on top of Prototypical Networks [9], learned by the nearest neighbor classifier on the semantic space measured by the Euclidean distance. We represent each class $y$ as a cluster and encode its categorical information by the cluster prototype $\mathbf{l}_y \in \mathbb{R}^D$:

$$\mathbf{l}_y = \frac{\sum_i \mathbf{x}_i \cdot \mathbb{1}_{[y_i = y]}}{\sum_i \mathbb{1}_{[y_i = y]}} \tag{1}$$

It should be noted that by "prototype" we mean the centroid of examples of a class. It should not be confused with the centroid of randomly sampled examples that is computed in each episode to train original Prototypical Networks.

We introduce translation mapping $R : \mathcal{Y}_n \mapsto \mathcal{P}(\mathcal{Y}_b)$ where $\mathcal{P}(\mathcal{Y}_b)$ is the powerset of the collection of all base classes. This defines a many-to-many relationship between novel and base classes, and

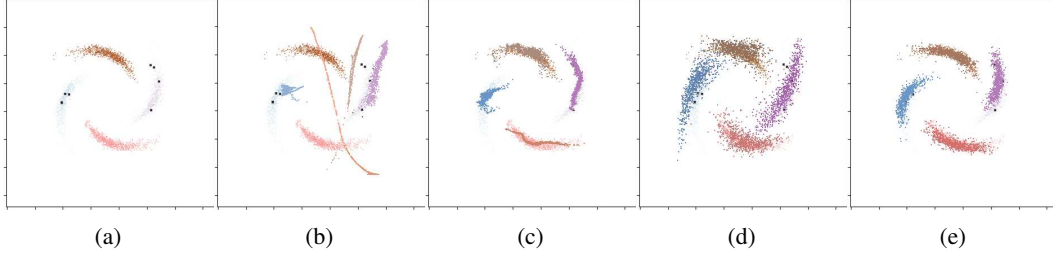

| (a) | (b) | (c) | (d) | (e) |

Figure 3: **Generation results on our toy dataset. (a):** Raw distribution of the "spiral" dataset, which consists of two base classes (top, bottom) and two novel classes (left, right). Novel classes are colored with lower saturation to indicate they are not available for training. Instead, only 4 examples (black crosses) are available. Generated samples are colored with higher saturation than the real data. **(b):** `c-GAN`; note that we also show the results of translating synthesized novel samples back to original base classes with another decoupled `c-GAN` for the visual consistency with the other variants; **(c):** `cCyc-GAN`; **(d):** `cDeLi-GAN`; **(e):** `cCov-GAN`. Results are best viewed in color with zoom.

is used to translate data from selected base classes to each novel class. To this end, given a novel class $y_n$, we compute its similarity scores $\alpha$ with all base classes $y_b$ using softmax over Euclidean distances between prototypes,

$$\alpha(y_b, y_n) = \frac{\exp\left(-\|\mathbf{l}_{y_b} - \mathbf{l}_{y_n}\|_2^2\right)}{\sum_{y_b' \in \mathcal{Y}_b} \exp\left(-\|\mathbf{l}_{y_b'} - \mathbf{l}_{y_n}\|_2^2\right)} \tag{2}$$

This results in a soft mapping (NBS-S) between base and novel classes, in which each novel class is paired with all base classes with soft scores. In practice, translating from all base classes is unnecessary, and computationally expensive. Alternatively, we consider a hard version of $R$ based on $k$-nearest neighbor search, where the top $k$ base classes are selected and treated as equal ($\alpha(y_b, y_n) = 1/k$). This hard mapping (NBS-H) saves memory, but introduces an extra hyper-parameter.

## 3.2 Adversarial Objective

After constraining our translation process to selected class pairs, we develop a baseline based on Conditional GAN (`c-GAN`) [15]. To this end, a discriminator $D_n$ is trained to classify real examples as the corresponding $N = |\mathcal{Y}_n|$ novel classes, and classify synthesized examples as an auxiliary "fake" class. [5]. The generator $G_n$ takes an example from base classes $R(y_n)$ that are paired with $y_n$ via NBS, and aims to fool the discriminator into classifying the generated example as $y_n$ instead of the "fake". More specifically, the adversarial objective can be written as:

$$\mathcal{L}_{\text{adv}}(G_n, D_n, \mathcal{B}, \mathcal{N}) = \mathbb{E}_{y_n \sim \mathcal{Y}_n}\left[\mathbb{E}_{\mathbf{x}_n \sim \mathcal{N}_{y_n}}\left[\log D_n(y_n|\mathbf{x}_n)\right]\right. \tag{3}$$

$$\left. + \mathbb{E}_{\mathbf{x}_b, y_b \sim \mathcal{B}_{R(y_n)}}\left[\alpha(y_b, y_n) \log D_n(N+1|G_n(y_n; \mathbf{x}_b, y_b))\right]\right] \tag{4}$$

where $\mathcal{N}_{y_n}$ consists of all novel examples labeled with $y_n$ in $\mathcal{N}$ while $\mathcal{B}_{R(y_n)}$ consists all base examples labeled by one of the classes in $R(y_n)$.

We train `c-GAN` by solving the minimax game of the adversarial loss. In this scenario, there is no explicit way to incorporate base classes intra-class variance into the generation of new novel examples. Also, any mappings that collapse synthesized features into existing observations yield the optimal solution [14]. These facts lead to unfavorable generation results as shown in Figure 3b. We next explore different ways to *explicitly* force the generator to learn the latent conditional distributions.

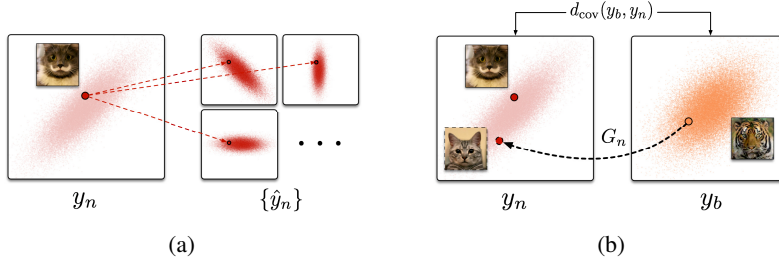

|  | | |
|---|---|---|
| $y_n$ | $\{\hat{y}_n\}$ | |
| (a) | | (b) |

Figure 4: **The importance of covariance in low-shot settings.** Suppose we have access to a single cat image during training. **(a):** conventional models can easily fail since there are infinite candidate distributions that cannot be discriminated; **(b):** related classes should have similar intra-class variances. Thus, we preserve covariance information during translation, to transfer knowledge from base classes to novel ones.

### 3.3 Cycle-consistency Objective

A natural idea for preventing modes from getting dropped is to apply the cycle-consistency constraint whose effectiveness has been proven over image-to-image translation tasks [16]. Besides extra supervision, it eliminates the demand for paired data, which is impossible to acquire for the low-shot learning setting. We extend this method for our conditional scenario and derive `cCyc-GAN`. Specifically, we learn two generators: $G_n$, which is our main target, and $G_b : \mathcal{N} \mapsto \mathcal{B}$ as an auxiliary mapping that reinforces $G_n$. We train the generators such that, the translation cycle recovers the original embedding in either a forward cycle $\mathcal{N} \mapsto \mathcal{B} \mapsto \mathcal{N}$ or a backward cycle $\mathcal{B} \mapsto \mathcal{N} \mapsto \mathcal{B}$. Our cycle-consistency objective could then be derived as,

$$\mathcal{L}_{\text{cyc}}(G_n, G_b) = \mathbb{E}_{y_n \sim \mathcal{Y}_n} \Bigg[ \mathbb{E}_{\mathbf{x}_n \sim \mathcal{N}_{y_n}, \mathbf{x}_b, y_b \sim \mathcal{B}_{R(y_n)}, \mathbf{z} \sim \mathcal{Z}} \, \alpha(y_b, y_n) \Bigg[ \tag{5}$$

$$\|G_n(y_n; G_b(y_b; \mathbf{x}_n, y_n, \mathbf{z}), y_b)\|_2^2 + \|G_b(y_b; G_n(y_n; \mathbf{x}_b, y_b), y_n, \mathbf{z})\|_2^2 \Bigg] \Bigg] \tag{6}$$

where a $Z$-dimensional noise vector sampled from a distribution $\mathcal{Z}$ is injected into $G_b$'s input since novel examples $x_n$ lack variability given the very limited amount of data. $\mathcal{Z}$ is a normal distribution $N(0, 1)$ for our `cCyc-GAN` model.

While $G_n$ is hard to train due to the extremely small data volume; $G_b$ has more to learn from, and can thus indirectly guide $G_n$ through its gradient. During our experiments, we found that cycle-consistency is indispensable for stabilizing the training procedure. Swaminathan *et al.* [25] observe that incorporating extra noise from a mixture of Gaussian distributions could result in more diverse results. Hence, we also report a variant called `cDeLi-GAN` which uses the same objective as `cCyc-GAN`, but sample the noise vector $\mathbf{z}$ from a mixture of $C$ different Gaussian distributions,

$$\mathcal{Z} \overset{d}{=} \frac{1}{C} \sum_{i=1}^{C} f(\mathbf{z}|\mu_i, \boldsymbol{\Sigma}_i), \quad \text{where } f(\mathbf{z}|\mu, \boldsymbol{\Sigma}) = \frac{\exp(-\frac{1}{2}(\mathbf{z} - \mu)^T \boldsymbol{\Sigma}^{-1}(\mathbf{z} - \mu))}{\sqrt{(2\pi)^Z |\boldsymbol{\Sigma}|}} \tag{7}$$

We follow the initialization setup in the previous work [25]. For each $\mu$, we sample from a uniform distribution $U(-1, 1)$. And for each $\boldsymbol{\Sigma}$, we first sample a vector $\sigma$ from a Gaussian distribution $N(0, 0.2)$, then we simply set $\boldsymbol{\Sigma} = \text{diag}(\sigma)$

Generation results of the two aforementioned methods are shown in Figure 3c and 3d. Both methods improve the diversity of generation compared to the naive `c-GAN`, yet they either under- or over-estimate the intra-class variance.

## 3.4 Covariance-preserving Objective

While cycle-consistency alone can transfer certain degrees of intra-class variance from base classes, we find it rather weak and unreliable since there are still infinite candidate distributions that cannot be discriminated based on limited observations (See Figure 4).

Building upon the assumption that similar classes share similar intra-class variance, one straightforward idea is to penalize the change of "variability" during translation. Hierarchical Bayesian models [23], prescribe each class as a multivariate Gaussian, where intra-class variability is embedded in a covariance matrix. We generalize this idea and try to maintain covariance in the translation process, although we model the class distribution by GAN instead of any prescribed distributions.

To compute the difference between two covariance matrices [26], one typical way is to measure the worst case distance between them using *Ky Fan m-norm*, *i.e.*, the sum of singular values of $m$-truncated SVD, which we denote as $\|[\cdot]_m\|_*$. To this end, we define the pseudo-prototype $\hat{\mathbf{l}}_{y_n}$ of each novel class $y_n$ as the centroid of all synthetic samples $\hat{\mathbf{x}}_n = G_n(y_n; \mathbf{x}_b, y_b)$ translated from related base classes. The covariance distance $d_{\mathrm{cov}}(y_b, y_n)$ between a base-novel class pair can then be formulated as,

$$d_{\mathrm{cov}}(y_b, y_n) = \left\|[\Sigma_{\mathbf{x}}(\mathbb{P}_{y_b}) - \Sigma_G(\mathbb{P}_{y_n})]_m\right\|_*, \quad \text{where} \begin{cases} \Sigma_{\mathbf{x}}(\mathbb{P}_y) = \frac{\sum_i (\mathbf{x}_i - \mathbf{l}_{y_i})(\mathbf{x}_i - \mathbf{l}_{y_i})^T \mathbb{1}_{[y_i = y]}}{\sum_i \mathbb{1}_{[y_i = y]}} \\ \Sigma_G(\mathbb{P}_y) = \frac{\sum_j (\hat{\mathbf{x}}_j - \hat{\mathbf{l}}_{y_j})(\hat{\mathbf{x}}_j - \hat{\mathbf{l}}_{y_j})^T \mathbb{1}_{[y_j = y]}}{\sum_j \mathbb{1}_{[y_j = y]}} \end{cases}$$

$$(8)$$

Consequently, our covariance-preserving objective can be written as the expectation of the weighted covariance distance using NBS-S,

$$\mathcal{L}_{\mathrm{cov}}(G_n) = \mathbb{E}_{y_n \sim \mathcal{Y}_n}\left[\mathbb{E}_{y_b \sim R(y_n)}\left[\alpha(y_b, y_n) d_{\mathrm{cov}}(y_b, y_n)\right]\right] \qquad (9)$$

Note that, for a matrix $\mathbf{X}$, $\|[\mathbf{X}]_m\|_*$ is non-differentiable with respect to itself, thus in practice, we calculate its subgradient instead. Specifically, we first compute the unitary matrices $\mathbf{U}$, $\mathbf{V}$ by $m$-truncated SVD [29], and then back-propagate $\mathbf{U}\mathbf{V}^T$ for sequential parameter updates. Proof of the correctness is provided in the supplementary material.

Finally, we propose our covariance-preserving conditional cycle-GAN, `cCov-GAN`, as:

$$G_n^* = \arg\min_{G_n, G_b} \max_{D_n, D_b} \mathcal{L}_{\mathrm{adv}}(G_n, D_n, \mathcal{B}, \mathcal{N})$$
$$+ \mathcal{L}_{\mathrm{adv}}(G_b, D_b, \mathcal{N}, \mathcal{B}) + \lambda_{\mathrm{cyc}}\mathcal{L}_{\mathrm{cyc}}(G_n, G_b) + \lambda_{\mathrm{cov}}\mathcal{L}_{\mathrm{cov}}(G_n) \qquad (10)$$

As illustrated in Figure 3e, preserving covariance information from relevant base classes to a novel class can improve low-shot generation quality. We attribute this empirical result to the interplay of adversarial learning, cycle consistency, and covariance preservation, that respectively lead to realistic generation, semantic consistency, and diversity.

## 3.5 Training

Following recent works on meta-learning [30, 11, 12], we design a two-stage training procedure. During the "meta-training" phase, we train our generative model with base examples only, by mimicking the low-shot scenario it would encounter later. After that, in the "meta-testing" phase, we are given novel classes as well as their low-shot examples. We use the trained $G_n$ to augment each class until it has the average capacity of the base classes. Then we train a classifier as one would normally do in a supervised setting using both real and synthesized data. For the choice of this final classifier, we apply the same one as in the original representation learning stage. For examples, we use the nearest neighbor classifier for embeddings from Prototypical Networks, and a normal linear classifier for those from ResNets.

We follow the episodic procedure used by [12] during meta-training. In each episode, we sample $N_b$ "meta-novel" classes from $\mathcal{B}$, and use the rest of $\mathcal{B}$ as "meta-base" classes. Then we sample $K_b$

examples from each meta-novel class as meta-novel examples. We compute the prototypes of each class and similarity scores between each "meta-novel" and "meta-base" class. To sample a batch of size $B$, we first include all "meta-novel" examples, and sample $B - N_b \cdot K_b$ examples uniformly from the "meta-base" classes retrieved by translation mapping $R$. Next, we push our samples through generations and discriminators to compute the loss. Finally, we update their weights for the current episode and start the next one.

## 4 Experiments

This section is organized as follows. In Section 4.1, we conduct low-shot learning experiments on the challenging ImageNet benchmark. In Section 4.2, we further discuss with ablation, both quantitatively and qualitatively, to better understand the performance gain. We demonstrate our model's capacity to generate diverse and reliable examples and its effectiveness in low-shot classification.

**Dataset** We evaluate our method on the real-world benchmark proposed by Hariharan *et al*. [11]. This is a challenging task because it requires us to learn a large variety of ImageNet [18] given a few exemplars for each novel classes. To this end, our model must be able to model the visual diversity of a wide range of categories and transfer knowledge between them without confusing unrelated classes. Following [11], we split the 1000 ImageNet classes into four disjoint class sets $\mathcal{Y}_b^{test}, \mathcal{Y}_n^{test}, \mathcal{Y}_b^{val}, \mathcal{Y}_n^{val}$, which consist of 193, 300, 196, 311 classes respectively. All of our parameter tuning is done on validation splits, while final results are reported using held-out test splits.

**Evaluation** We repeat sampling novel examples five times for held-out novel sets and report results of mean top-5 accuracy in both *conventional* low-shot learning (LSL, to test on novel classes only) and its *generalized* setting (GLSL, to test on all categories including base classes).

**Baselines** We compare our results to the exact numbers reported by Feature Hallucination [11] and Feature Imagination [12]. We also compared to other non-generative methods including classical Siamese Networks [31], Prototypical Networks [9], Matching Networks [8], and MAML [32] as well as more recent Prototypical Matching Networks [12] and Attentive Weight Generators [33]. For stricter comparison, we provide two extra baselines to exclude the bias induced by different embedding methods: P-FH builds on Feature Hallucinating by substituting their non-episodic representation with learned prototypical features. Another baseline (first row in Table 1), on the contrary, replaces prototypical features with raw ResNet-10 embeddings. The results for MAML and SN are reported using their published codebases online.

**Implementation details** Our implementation is based on PyTorch [34]. Since deeper networks would unsurprisingly result in better performance, we confine all experiments in a ResNet-10 backbone[1] with a 512-d output layer. We fine-tune the backbone following the procedure described in [11]. For all generators, we use three-layer MLPs with all hidden layers' dimensions fixed at 512 as well as their output for synthesized features. Our discriminators are accordingly designed as three-layer MLPs to predict probabilities over target classes plus an extra fake category. We use leaky ReLU of slope 0.1 without batch normalization. Our GAN models are trained for 100000 episodes by ADAM [35] with initial learning rate fixed at 0.0001 which anneals by 0.5 every 20000 episodes. We fix the hyper-parameter $m = 10$ for computing truncated SVD. For loss term contributions, we set $\lambda_{\text{cyc}} = 5$ and $\lambda_{\text{cov}} = 0.5$ for all final objectives. We choose $Z = 100$ as the dimension of noise vectors for $G_b$'s input, and $C = 50$ for the Gaussian mixture. We inject prototype embeddings instead of one-hot vectors as categorical information for all networks (prototypes for novel classes are computed using the low-shot examples only). We empirically set batch size $B = 1000$, and $N_b = 20$ and $K_b = 10$ for all training, no matter what would the number of shots be in the test. This is more efficient but possibly less accurate than [9] who trained separate models for each testing scenario, so the number of shots in train and test always match. All hyper-parameters are cross-validated on the validation set using a coarse grid search.

### 4.1 Main Results

For comparisons, we include numbers reported in previous works under the same experimental settings. Note that the results for MAML and SN are reported using their published codebases

Table 1: Low-shot classification top-5 accuracy% of all comparing methods under LSL and GLSL settings on ImageNet dataset. All results are averaged over five trials separately, and omit standard deviation for all numbers are of the order of $0.1\%$. The **best** and second best methods under each setting are marked in according formats.

| Method | Representation | Generation | LSL $K=1$ | 2 | 5 | 10 | 20 | GLSL $K=1$ | 2 | 5 | 10 | 20 |
|---|---|---|---|---|---|---|---|---|---|---|---|---|
| Baseline | ResNet-10 [36] | - | 38.5 | 51.2 | 64.7 | 71.6 | 76.3 | 40.6 | 49.8 | 64.3 | 72.1 | 76.7 |
| | SN [31] | - | 38.9 | - | 64.6 | - | 76.4 | 48.7 | - | 68.3 | - | 73.8 |
| | MAML [32] | - | 39.2 | - | 64.2 | - | 76.8 | 49.5 | - | 69.6 | - | 74.2 |
| | PN [9] | - | 39.4 | 52.2 | 66.6 | 72.0 | 76.5 | 49.3 | 61.0 | 69.6 | 72.8 | 74.7 |
| | MN [8] | - | 43.6 | 54.0 | 66.0 | 72.5 | 76.9 | 54.4 | 61.0 | 69.0 | 73.7 | 76.5 |
| | PMN [12] | - | 43.3 | 55.7 | 68.4 | 74.0 | 77.0 | 55.8 | 63.1 | 71.1 | 75.0 | 77.1 |
| | AWG [33] | - | 46.0 | 57.5 | 69.2 | 74.8 | 78.1 | 58.2 | 65.2 | 72.7 | 76.5 | 78.7 |
| FH [11] | ResNet-10 | LR w/ A. | 40.7 | 50.8 | 62.0 | 69.3 | 76.4 | 52.2 | 59.7 | 68.6 | 73.3 | 76.9 |
| P-FH | PN | LR w/ A. | 41.5 | 52.2 | 63.5 | 71.8 | 76.4 | 53.6 | 61.7 | 69.0 | 73.5 | 75.9 |
| FI [12] | PN | meta-learned LR | 45.0 | 55.9 | 67.3 | 73.0 | 76.5 | 56.9 | 63.2 | 70.6 | 74.5 | 76.5 |
| | PMN | meta-learned LR | 45.8 | 57.8 | 69.0 | 74.3 | 77.4 | 57.6 | 64.7 | 71.9 | 75.2 | 77.5 |
| CP-AAN | ResNet-10 | cCov-GAN | 47.1 | 57.9 | 68.9 | 76.0 | 79.3 | 52.1 | 60.3 | 69.2 | 72.4 | 76.8 |
| (Ours) | PN | c-GAN | 38.6 | 51.8 | 64.9 | 71.9 | 76.2 | 49.4 | 61.5 | 69.7 | 73.0 | 75.1 |
| | PN | cCyc-GAN | 42.5 | 54.6 | 66.7 | 74.3 | 76.8 | 57.6 | 65.1 | 72.2 | 73.9 | 76.0 |
| | PN | cDeLi-GAN | 46.0 | 58.1 | 68.8 | 74.6 | 77.4 | 58.0 | 65.1 | 72.4 | 74.8 | 76.9 |
| | PN | cCov-GAN | 48.4 | 59.3 | 70.2 | 76.5 | 79.3 | 58.5 | 65.8 | 73.5 | 76.0 | 78.1 |

LR w/ A.: Logistic Regressor with Analogies.

online. We decompose each method into stage-wise operations for breaking performance gain down to detailed choices made in each stage.

We provide four models constructed with different GAN choices as justified in Section 3. All of our introduced CP-AAN approaches are trained with NBS-S which would be further investigated with ablation in the next subsection. Results are shown in Table 1. Our best method consistently achieves significant improvement over the previous augmentation-based approaches for different values of $K$ under both LSL and GLSL settings, achieving almost $2\%$ performance gain compared to baselines. We also notice that apart from overall improvement, our best model achieves its largest boost (~9%) at the lowest shot over naive baseline and $2.6\%$ over Feature Imagination (FI) [12] under the LSL setting, even though we use a simpler embedding technique (PN compared to their PMN). We believe such performance gain can be attributed to our advanced generation methods since at low shots, FI applies *discrete* transformations that its generator has previously learned while we can now sample through a *smooth* distribution combining all related base classes' covariance information.

Note that in the LSL setting, all generative methods assume we still have access to original base examples when learning final classifiers while non-generative baselines usually don't have this constraint.

## 4.2 Discussions

In this subsection, we carefully examine our design choices for the final version of our CP-AAN. We start by unpacking performance gain over the standard batch sampling procedure and proceed by showing both quantitative and qualitative evaluations on generation quality.

**Ablation on NBS**    To validate the effectiveness of the NBS strategy over standard batch sampling for feature augmentation, we conduct an ablation study to show our absolute performance gain in Figure 5a. In general, we empirically demonstrate that applying NBS improves the performance of low-shot recognition. We also show that the performance of NBS-H is sensitive to the hyper-parameter $k$ in the $k$-nearest neighbor search. Therefore, the soft assignment is preferable if computational resources allow.

**Quantitative Generation Quality**    We next quantitatively evaluate the generation quality of the variants introduced in Section 3 and previous works as shown in Figure 5b. Note that for FH, we used their published codebase online; for FI, we implemented the network and train with the procedure described in the original paper. We measure the diversity of generation via the mean average pairwise Euclidean distance of generated examples within each novel class. We adopt same augmentation strategies as used for ImageNet experiments. For reference, the mean average Euclidean distance over real examples is $0.163$. In summary, the results are consistent with our expectation and support

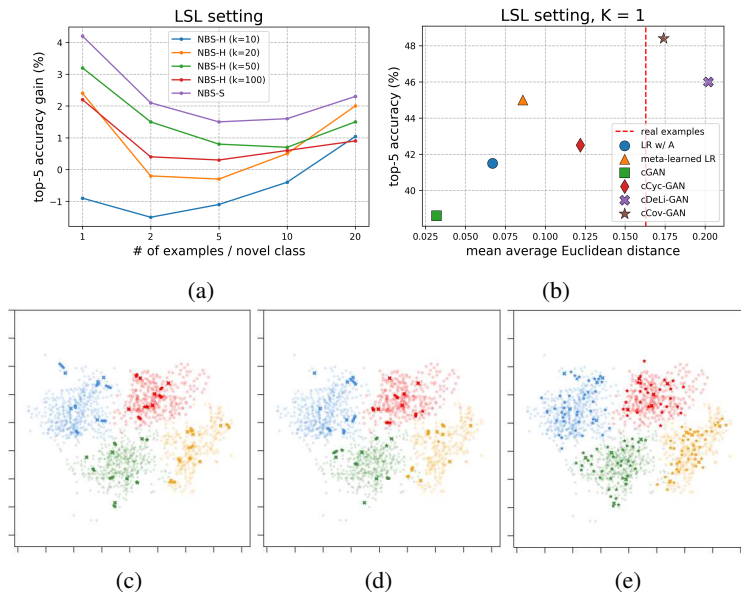

(a)                                                      (b)

(c)                          (d)                          (e)

Figure 5: **Ablation analysis.** **(a):** Unpacked performance gain for each NBS strategy; **(b):** accuracy vs. diversity; **(c, d):** Feature Hallucination and Feature Imagination lack diverse modes; **(e):** our best method could synthesize both diverse and realistic embeddings. Results are best viewed in color with zoom.

our design choices. Feature Hallucination and Imagination show less diversity compared to real data. Naive `c-GAN` even under-performs those baselines due to the mode collapse. Cycle-consistency and Gaussian mixture noise do help generation in both accuracy and diversity. However, they either under- or over-estimate the diversity. Our covariance-preserving objective leads to the best hallucination quality, since the generated distribution more closely resembles the real data diversity. Another insight from Figure 5b is that not surprisingly, under-estimating data diversity is more detrimental to classification accuracy than over-estimating.

**Qualitative Generation Quality**     Figure 5c, 5d, 5e show t-SNE [37] visualizations of the data generated by Feature Hallucination, Feature Imagination and our best model in the prototypical feature space. We fix the number of examples per novel class $K = 5$ in all cases and plot their real distribution with translucent point clouds. The 5 real examples are plotted in crosses and synthesized examples are denoted by stars. Evidently, naive generators could only synthesize novel examples that are largely pulled together. Although t-SNE might visually drag similar high dimensional points towards one mode, our model shows more diverse generation results that are better aligned with the latent distribution, improving overall recognition performance by spreading seed examples in meaningful directions.

## 5   Conclusion

In this paper, we have presented a novel approach to low-shot learning that augments data for novel classes by training a cyclic GAN model, while shaping intra-class variability through similar base classes. We introduced and compared several GAN variants in a logical process and demonstrated the increasing performance of each model variant. Our proposed model significantly outperforms the state of the art on the challenging ImageNet benchmark in various settings. Quantitative and qualitative evaluations show the effectiveness of our method in generating realistic and diverse data for low-shot learning, given very few examples.

**Acknowledgments** This work was supported by the U.S. DARPA AIDA Program No. FA8750-18-2-0014. The views and conclusions contained in this document are those of the authors and should not be interpreted as representing the official policies, either expressed or implied, of the U.S. Government. The U.S. Government is authorized to reproduce and distribute reprints for Government purposes notwithstanding any copyright notation here on.

## Footnotes

[1]Released on `https://github.com/facebookresearch/low-shot-shrink-hallucinate`

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
