[Supplementary Material · cpaan_neurips19_camera_ready_supplement.pdf]

## A  Details about Neighborhood Batch Sampling

In this section, we cover more details in regard to Neighborhood Batch Sampling (NBS). We have considered two instantiations of the translation mapping $R$ and similarity scores $\alpha$, based on hard $k$-nearest neighbor search and soft selection, respectively. Given a novel class $y_n$, we want to select the base classes $\{y_b\}$ that are semantically similar to the $y_n$ query.

**Hard assignments (NBS-H)** This sampling method retrieves $k$ uniformly weighted nearest base classes. NBS-H can be formulated as follows,

$$R(y_n) = \underset{\mathcal{Y}_b' \subset \mathcal{Y}_b, |\mathcal{Y}_b'|=k}{\arg\min} \sum_{y_b \in \mathcal{Y}_b'} \|\mathbf{l}_{y_b} - \mathbf{l}_{y_n}\|_2^2, \quad \alpha(y_b, y_n) = \frac{1}{k} \, \forall y_b \in R(y_n). \tag{11}$$

Similar heuristics are used in previous works [11, 12] as well by introducing a new hyper-parameter $k$. Though NBS-H may save computational resources, in practice, we find it too sensitive to the selection of $k$. In addition to that, it treats all selected base classes as equally related to the target novel class $y_n$, which slows the convergence and hurts the performance.

**Soft assignments (NBS-S)** In this case, all base classes are considered, and weighted by the softmax score over the learned metrics,

$$R(y_n) = \mathcal{Y}_b, \quad \alpha(y_b, y_n) = \frac{\exp\left(-\|\mathbf{l}_{y_b} - \mathbf{l}_{y_n}\|_2^2\right)}{\sum_{y_b' \in \mathcal{Y}_b} \exp\left(-\|\mathbf{l}_{y_b'} - \mathbf{l}_{y_n}\|_2^2\right)}. \tag{12}$$

Through the ablation study, we showed that this batch sampling technique is more effective than NBS-H given enough computational resources.

## B  Details about Intermediate GAN Objectives

In this section, we formulate our full objectives for intermediate variants derived for the imbalanced set-to-set translation.

c-GAN    Its full objective could be defined as a basic minimax game,

$$G_n^* = \arg\min_{G_n} \max_{D_n} \mathcal{L}_{\text{adv}}(G_n, D_n, \mathcal{B}, \mathcal{N}). \tag{13}$$

cCyc-GAN    Accordingly, its full objective can be directly derived from cycle-consistency,

$$G_n^* = \arg\min_{G_n, G_b} \max_{D_n, D_b} \mathcal{L}_{\text{adv}}(G_n, D_n, \mathcal{B}, \mathcal{N}) + \mathcal{L}_{\text{adv}}(G_b, D_b, \mathcal{N}, \mathcal{B}) + \lambda_{\text{cyc}} \mathcal{L}_{\text{cyc}}(G_n, G_b). \tag{14}$$

## C  Details about Computing Subgradient of Ky Fan $m$-norm

**Theorem 1**  *Given a matrix $\mathbf{X}$ and its Ky Fan $m$-norm $\|[\mathbf{X}]_m\|_* = \sum_i \sigma_i(\tilde{\mathbf{X}})$ where $\tilde{\mathbf{X}} = \mathbf{U}\boldsymbol{\Sigma}\mathbf{V}^T$ is the $m$-truncated SVD and $\sigma_i(\cdot)$ is the $i$-th largest singular value, we have,*

$$\frac{d\|[\mathbf{X}]_m\|_*}{d\mathbf{X}} = \mathbf{U}\mathbf{V}^T \tag{15}$$

**Proof**  Rewrite Ky Fan $m$-norm by its sub-differential set,

$$\|[\mathbf{X}]\|_* = \text{tr}(\boldsymbol{\Sigma}) = \text{tr}(\boldsymbol{\Sigma}\boldsymbol{\Sigma}^{-1}\boldsymbol{\Sigma}) \tag{16}$$

Then,

$$d\|[\mathbf{X}]_m\|_* = \text{tr}(\boldsymbol{\Sigma}\boldsymbol{\Sigma}^{-1}d\boldsymbol{\Sigma}) \tag{17}$$

Since we have,

$$\mathrm{d}\mathbf{X} = \mathrm{d}\mathbf{U}\boldsymbol{\Sigma}\mathbf{V}^T + \mathbf{U}\mathrm{d}\boldsymbol{\Sigma}\mathbf{V}^T + \mathbf{U}\boldsymbol{\Sigma}\mathrm{d}\mathbf{V}^T \tag{18}$$

Therefore,

$$\mathbf{U}\mathrm{d}\boldsymbol{\Sigma}\mathbf{V}^T = \mathrm{d}\mathbf{X} - \mathrm{d}\mathbf{U}\boldsymbol{\Sigma}\mathbf{V}^T - \mathbf{U}\boldsymbol{\Sigma}\mathrm{d}\mathbf{V}^T$$
$$\Rightarrow \mathrm{d}\boldsymbol{\Sigma} = \mathbf{U}^T\mathrm{d}\mathbf{X}\mathbf{V} - \mathbf{U}^T\mathrm{d}\mathbf{U}\boldsymbol{\Sigma} - \boldsymbol{\Sigma}\mathrm{d}\mathbf{V}^T\mathbf{V} \tag{19}$$

By the diagonality of $\boldsymbol{\Sigma}$ and anti-symmetricity of $\mathbf{U}$, $\mathbf{V}$,

$$\mathbf{U}^T\mathrm{d}\mathbf{U}\boldsymbol{\Sigma} + \boldsymbol{\Sigma}\mathrm{d}\mathbf{V}^T\mathbf{V} = 0$$
$$\Rightarrow \mathrm{d}\boldsymbol{\Sigma} = \mathbf{U}^T\mathrm{d}\mathbf{X}\mathbf{V} \tag{20}$$

Substitute it into Equation 17,

$$\mathrm{d}\|[\mathbf{X}]_m\|_* = \mathrm{tr}(\boldsymbol{\Sigma}\boldsymbol{\Sigma}^{-1}\mathrm{d}\boldsymbol{\Sigma}) = \mathrm{tr}(\mathbf{U}^T\mathrm{d}\mathbf{X}\mathbf{V}) = \mathrm{tr}(\mathbf{U}^T\mathbf{V}\mathrm{d}\mathbf{X})$$
$$\Rightarrow \frac{\mathrm{d}\|[\mathbf{X}]_m\|_*}{\mathrm{d}\mathbf{X}} = \mathbf{U}\mathbf{V}^T \tag{21}$$

$\square$