[Reviews · NeurIPS 2018]

Reviewer 1



*** After rebuttal: Thank you for the additional experiments and clarifications, they were very helpful. I feel that both the precise training algorithm as well as the distinction of the two types of prototypes are important points to add to the revised version. I agree with the other reviewers that Section 3 can be written in a more clear way, and it would also be helpful to double-check the text for grammar for the final version. *** Original review: This paper put forward a novel approach to low-shot learning by feature augmentation. The problem of low-shot learning is to learn classifiers between a set of previously-unseen classes given only a few examples of each. It is assumed that a potentially large set of ‘training’ or ‘base’ classes is given, but this set is disjoint from the set of classes that will be presented at test time. Given the insufficiency of the training data for learning a low-shot classifier, generating additional data is a reasonable strategy and there has been increasingly more research on this recently (the authors correctly cite the relevant works). The approach presented here works by ‘translating’ examples of the base classes into examples of novel classes, inspired by unsupervised set-to-set translation methods. This method is designed to overcome a weakness of previous work with regards to the variance of these generated examples. In particular, since the true distribution of the novel class is not known, it’s difficult to estimate its intra-class variance. Previous works have implicitly assumed that all classes have the same variance, which is an unrealistic assumption. This work tackles this problem by instead assuming that a novel class will have similar covariance to the covariance of its most related base classes. In every low-shot learning episode, comprised of a set of N novel classes, they sample a set of base classes that are relevant to those novel classes. The relevance between a novel and base classes is determined via a metric inspired by the prototypical network approach: a prototype is created for each base class, which is the average representation of its examples, and a prototype is created for each novel class similarly. The base class prototypes that are the nearest-neighbors of a novel class prototype correspond to its most related classes. In every such batch consisting of N novel classes and the set of most relevant base classes, an adversarial network is employed to ‘transform’ each base class example into a novel class example. The GAN is trained as usual with the difference that the generator’s classification task has (N+1) targets: each example will either belong to one of the N novel classes of the batch, or is a generated example (ie an example that is transformed from one of the related base classes into something that should look like an example of a novel class). Successful training of this GAN will make the generated examples of the novel classes indistinguishable from the real examples of the novel classes. They then introduce additional components to the loss function to match the covariance of the generated examples for a class to the covariance of the real class from which they were transformed. Pros - Using the intra-class variance of classes that are related to the novel class seems like a reasonable proxy for the intra-class variance of the novel class. Generating examples this way seems indeed like a promising way of sampling more diverse examples that are still plausible representatives of the novel class. - They progressively add layers of complication to their method. By evaluating each of these variants they justify that their final approach is well-motivated and all its components are necessary for achieving good performance. The simplest version is a standard conditional GAN whose purpose is to translate examples of base classes into examples of novel classes such that these generated examples are indistinguishable from the real examples of novel classes. Then, they take extra steps to ensure that the distribution of the base class that is most related to a novel class is faithfully maintained during translation. They do this using a cyclized GAN objective. Their final variant adds an additional component to the loss that explicitly penalizes differences in the covariance between the related base class and the translated examples for a given novel class. - They verify both quantitatively and qualitatively that their generated images are more diverse than competing approaches - Their method achieves state-of-the-art results on ImageNet Cons - An important component of the algorithm was left unclear: how the training progresses. One potential option would be to train episodically, through a series of training episodes each of which mimic a low-shot test episode. In that case, N base classes would be sampled for a training episode, and their most related other base classes would be included (as determined by class prototypes) would contribute the examples that would be ‘translated’ into the N chosen base classes. Then a classifier for this episode would be learned using the k real examples of each of the N classes, together with the fake examples. Finally, based on the performance of this newly learned classifier on a disjoint set of examples originating from these N classes, the model’s weights are updated (e.g. embedding network, generator parameters, etc). Though this seems natural, it’s not stated whether this is indeed how training proceeds. It’s important to clarify this point. - Clarity is sometimes compromised due to using certain words incorrectly. The word ‘incur’ is used multiple times in a confusing way for example. - Experimentally it would be nice to compare to other low-shot learning methods that do not necessarily work by feature augmentation. For example metric learning and meta-learning methods present a different solution which is shown to be effective as well. It would be nice to compare against some representatives of these families, e.g. Prototypical Network, Matching Network, Siamese Network, MAML (Finn et al, 2017). Other comments: - It should be noted that this approach utilizes an additional piece of information than many of the competitors: it utilizes the data of the base classes not only at meta-training time but also during meta-testing. In particular, even at test time the real data of the base classes constitutes the input to the generator for translating those images into fake data of the novel classes. - A prototypical network is an episodic model: it averages the representations of the points *of a given episode* that belong to the same class to form the prototype of that class. Here, as I understand it, to form the prototype of a class, all examples in the entire base set that belong to that class are averaged. Therefore I wouldn’t call this a Prototypical Network. This process can just be explained as representing each class as a cluster whose mean is the average of the features of the corresponding class points. - Prototypical Matching Network (PMN) is an entry in the table, without a reference. Also, no description is provided for that method. Please cite the appropriate work. - Similarly for P-FH: there is a short description provided but unclear which previous work this method comes from. Is this a new proposed baseline? If so it would be good to give more details. - Line 248: “its embedding technique is more complex”, referring to the embedding technique of FI compared to this proposed model. It wasn’t clear why it’s more complex. In a nutshell, the authors put forth a new model for low-shot learning by feature augmentation whose purpose is to generate as diverse as possible examples that still plausibly belong to each novel class. They experimentally demonstrate that their approach achieves state-of-the-art on ImageNet and generated more diverse examples than previous works. The down sides of the paper are mostly in terms of clarity, caused both by the inappropriate or confusing use of words as well as omitting details of the algorithm that I feel are important. Finally, it would be useful to compare experimentally against other approaches that tackle low-shot learning in ways other than feature augmentation.

Reviewer 2



This paper proposes an approach to tackle the problem of low-shot learning with a feature augmentation method using GANs. It builds upon previous feature augmentation approaches by using a generative model of the latent feature distribution for novel classes and selectively borrows from the known bases classes using weights based on inter-class distances in the learned metric space. The proposed GAN incorporates cycle-consistency and covariance matching to encourage the features generated for the new classes to generalize those of the base classes in a useful way. Experiments on low-shot ImageNet demonstrate a noticeable increase in accuracy relative to previous methods. Pros - Approach is novel and incorporates ideas from generative modeling into feature augmentation in an interesting way. - Results clearly show relative contribution of each proposed component. Cons - Clarity is low in several places (most prominently in Section 3). - Comparison to many recent few-shot learning methods on e.g. miniImageNet is lacking. The clarity of the paper is OK but there are some sections should be made more precise. Section 3 in particular has undefined expressions and refers to the supplementary material too much to be self-contained. It was unclear to me whether the generator takes as input a noise vector. The notation G_n(y_n; x_b, y_b) suggests that there is no noise as input, but the discussion about a mixture of Gaussian noise distribution in lines 160-163 suggests that there is. Also in Section 4.1, the four disjoint shattered sets discussed in lines 225-227 is not entirely clear. I would recommend explaining this in greater detail. The equations seem to be correct to me, but I did not attempt to verify the subgradient of the Ky Fan m-norm in the supplementary materials. Overall, the paper explores several intuitively motivated concepts for improving low-shot learning and demonstrates good performance gains on the standard benchmark for this task. Minor issues: - Line 1: grammar, attacks the low-shot learning problem - Line 2: grammar, in the low data regime - Line 77: consider changing incur to another word e.g. impart - Line 78: grammar, into their novel counterparts - Line 79: typo, computational - Line 83-84: grammar, from a handful of observationss - Line 86: grammar, Bayesian methods - Line 88: typo, Literature - Line 93: consider changing incurring to another word e.g. applying - Line 98-99: grammar, rather than certain - Line 115: grammar, requires a meaningful - Line 123: grammar, the current batch - Line 130-131: grammar, For more details - Line 149: typo, proven - Line 151: grammar, that further constrains - Line 152-153: grammar, the translation cycle should be able to recover the original embedding - Line 213-214: grammar, after training the whole network - Line 218: typo, decays - Line 225: grammar, The original benchmark ===== After Author Feedback ===== I thank the authors for providing feedback -- I maintain my original recommendation.

Reviewer 3



This paper considers the problem of few-shot learning by exploring ways in which data can be generated in order to augment small datasets so that the model can generalize better. It considers feature augmentation, avoiding the unnecessary complexity involved with generating raw data (in this case, images). The setup is the following: we have a base set of classes (consisting of a lot of examples) and a novel set of classes (consisting of a small number of examples) and the goal is to learn to generate data for a novel class based on data from the base classes. Though there has been previous work on feature augmentation in this kind of setup (Hariharan 2017; Wang 2018), the paper’s novelty is that it does not use all base classes equally to generate new data for a novel class but attempts to use only base classes whose semantics and statistics match the novel class. The paper’s main contribution is to formulate a GAN objective that facilitates such feature augmentation. In addition to the normal conditional GAN objective (which is supplemented by a relation mapping which only considers closest base classes to a novel class), the following extensions are added to improve feature generation (both of which help provide similar intra-class variance between related base and novel classes): Cycle-consistency objective Covariance-preserving objective Experiments are conducted on ImageNet low-shot learning benchmark used in previous work, and performance improvement is shown over previous feature generation work in a variety of setups. Ablation studies are done to show that each addition to the GAN objective benefits performance. Pros Figures were very useful in understanding how method is working. Figure 3 especially helps determine the benefit of each part in the objective. The experiments are thorough and the ablation studies also demonstrate the utility of each suggested addition to the GAN objective. Cons Method seems to have a lot of different parts and associated hyperparameters (with respect to weighting GAN objectives and within GAN training of each objective) which may make results hard to recreate. Comments Typo @ line 144: “into exsted observations” => “into existing observations” It seemed to me that some experimental details were missing (specifically, how model is trained in the low-shot learning framework for ImageNet) and I had to refer to Hariharan 2017 for those details. Might be nice to include a short description of this procedure to make the paper more self-contained. At first I found Figure 3 confusing as to why generated data was being visualized for the base classes also. I believe this is to show the benefit of cycle-consistency objective but might be good to make a note as to why generated data for base classes is also shown when we are only interested in generated data for novel classes. Hariharan, Bharath et al. Low-shot Visual Recognition by Shrinking and Hallucinating Features. 2017. Wang, Yu-Xiong et al. Low-Shot Learning from Imaginary Data. 2018.